# Comparative Study of Constipation Exacerbation by Potassium Binders Using a Loperamide-Induced Constipation Model

**DOI:** 10.3390/ijms21072491

**Published:** 2020-04-03

**Authors:** Yuki Narita, Koichi Fukumoto, Masaki Fukunaga, Yuki Kondo, Yoichi Ishitsuka, Hirofumi Jono, Tetsumi Irie, Hideyuki Saito, Daisuke Kadowaki, Sumio Hirata

**Affiliations:** 1Department of Pharmacy, Kumamoto University Hospital, 1-1-1 Honjo, Chuo-ku, Kumamoto 860-8556, Japan; 2Department of Clinical Pharmaceutical Sciences, Graduate School of Pharmaceutical Sciences, Kumamoto University, 1-1-1 Honjo, Chuo-ku, Kumamoto 860-8556, Japan; 3Department of Clinical Pharmacology, Graduate School of Pharmaceutical Sciences, Kumamoto University, 5–1 Oe-honmachi, Chuo-ku, Kumamoto 862-0973, Japan; 4Department of Clinical Chemistry and Informatics, Graduate School of Pharmaceutical Sciences, Kumamoto University, 5–1 Oe-honmachi, Chuo-ku, Kumamoto 862-0973, Japan; 5Department of Clinical Pharmaceutics, Faculty of Pharmaceutical Sciences, Sojo University, 4-22-1 Ikeda, Nishi-ku, Kumamoto 860-0082, Japan

**Keywords:** hemodialysis, constipation, rat, calcium polystyrene sulfonate, sodium polystyrene sulfonate

## Abstract

Patients on dialysis are frequently administered high doses of potassium binders such as calcium polystyrene sulfonate (CPS) and sodium polystyrene sulfonate (SPS), which exacerbate constipation. Here, we compare the degree of constipation induced by CPS and SPS using a loperamide-induced constipation model to identify the safer potassium binder. Constipation model was created by twice-daily intraperitoneal administration (ip) of loperamide hydrochloride (Lop; 1 mg/kg body weight) in rats for 3 days. Rats were assigned to a control group, Lop group, Lop + CPS group or Lop + SPS group, and a crossover comparative study was performed. Defecation status (number of feces, feces wet weight, fecal water content and gastrointestinal transit time (GTT)) was evaluated. In the Lop + CPS group, GTT was significantly longer, and fecal water content was reduced. In the Lop + SPS group—although the fecal water content and GTT were unaffected—the number of fecal pellets and the fecal wet weight improved. Thus, SPS was less likely to cause constipation exacerbation than CPS. Considering the high frequency of constipation in dialysis patients with hyperkalemia, preferentially administering SPS over CPS may prevent constipation exacerbation.

## 1. Introduction

Patients on dialysis are prone to constipation due to age-related intestinal dysmotility, insufficient dietary fiber intake due to potassium restriction and restricted water intake. Intestinal dysmotility disorders are the leading cause of constipation in dialysis patients [1], which over 50% experience constipation [2,3,4,5]. Furthermore, patients on dialysis have reduced potassium excretion due to impaired renal function and are thus, prone to hyperkalemia [6,7,8]. Elevated serum potassium may cause sudden death, and hence, correction of hyperkalemia is critical [6,9,10,11,12]. Therefore, positive ion-exchange resins, such as calcium polystyrene sulfonate (CPS) and sodium polystyrene sulfonate (SPS), which act as potassium binders, are used in many patients. These resins decrease the serum potassium by absorbing it through the ion-exchange in the lumen of the distal large intestine and promoting potassium excretion in the feces [13,14,15]. CPS and SPS are water-insoluble ion-exchange resins, typically administered at high oral doses of 15 to 30 g/day in Japan and similarly high doses of 15 to 60 g/day in the United States and Canada. Therefore, although the large intestine promotes smooth passage of feces by peristalsis and mucus secretion, when high doses of CPS and SPS cause obstruction, the increased colon water absorption results in fecal impaction, exacerbating the obstruction. This further worsens constipation in the dialysis patients, which—if not managed successfully with appropriate laxatives—can cause fatal intestinal diseases, such as intestinal obstruction [16] and gastrointestinal perforation [17,18]. Consequently, countermeasures against constipation caused by potassium binders in dialysis patients are extremely important, and it is necessary to determine if CPS or SPS is safer and less likely to exacerbate constipation.

In this study, we compared the degree of constipation induced by CPS and SPS using a loperamide-induced constipation model reflecting intestinal dysmotility, which is a major cause of constipation in dialysis patients, to identify a safer potassium binder concerning constipation.

## 2. Results

### 2.1. Effects of Calcium Polystyrene Sulfonate (CPS) or Sodium Polystyrene Sulfonate (SPS) on Body Weight, Food Intake, Water Intake and Urine Volume

Body weight, food intake, water intake and urinary volume data for all groups are shown in Table 1. All the parameters, including food and water intake, which are external factors that affect constipation, were not significantly different among the four groups.

### 2.2. Effects of CPS or SPS on Serum Electrolytes

Serum sodium (Na), potassium (K) and chloride (Cl) concentrations are shown in Table 2. No electrolyte imbalance was observed in the loperamide-induced constipation model, and CPS or SPS did not affect electrolyte concentration.

### 2.3. Effect of CPS or SPS on the Number of Fecal Pellets

The total number of fecal pellets is shown in Figure 1a; the cumulative number of fecal pellets over time is shown in Figure 1b. The number of fecal pellets is one of the parameters for evaluating constipation, which reduces with constipation [19]. The total number of fecal pellets was significantly reduced in the Lop group compared to that in the control group and similarly decreased in the Lop + CPS group (Control: 46.4 ± 10.7 pellets/18 h, Lop: 29.9 ± 7.7 pellets/18 h, Lop + CPS: 29.6 ± 7.1 pellets/18 h). However, the total number of fecal pellets was higher in the Lop + SPS group (42.5 ± 11.2 pellets/18 h) than that in the Lop group and was improved to the same degree as the control group. The cumulative number of fecal pellets over time was also the same, with a decrease in the Lop group and the Lop + CPS group compared to that in the control group and an increase in the Lop + SPS group to about the same value as that in the control group.

### 2.4. Effect of CPS or SPS on Fecal Wet Weight

The total fecal wet weight is shown in Figure 2a; the cumulative fecal wet weight over time is shown in Figure 2b. The total fecal wet weight was significantly reduced in the Lop group compared to that in the control group (Control: 11.9 ± 2.2 g/18 h, Lop: 6.5 ± 2.0 g/18 h). The cumulative fecal wet weight in the Lop + CPS group (8.8 ± 2.1 g/18 h) also decreased compared to that in the control group, but there was no significant difference. On the other hand, the total fecal wet weight was heavier in the Lop + SPS group (14.8 ± 3.5 g/18 h) than that in the Lop group and was improved to the same degree as the control group. In addition, no visual diarrhea was observed in the Lop + SPS group. The cumulative fecal wet weight over time was also the same, with a decrease in the Lop group and the Lop + CPS group compared to that in the control group and an increase in the Lop + SPS group to approximately the same value as that in the control group.

### 2.5. Effect of CPS or SPS on Fecal Water Content

Fecal water content is shown in Figure 3. Fecal water content was similar between the control group and the Lop group (Control: 64.1% ± 4.3%, Lop: 58.2% ± 4.6%). However, fecal water content was significantly lower in the Lop + CPS group (46.8% ± 5.6%; *p* < 0.01) than the Lop group. In contrast, the Lop + SPS group (60.6% ± 3.2%) was comparable to the Lop group, with no significant differences. Moreover, the fecal water content in the Lop + CPS group was significantly lower than that in the Lop + SPS group (*p* < 0.01).

### 2.6. Effect of CPS or SPS on Gastrointestinal Transit Time (GTT)

Gastrointestinal transit time (GTT) is shown in Figure 4. GTT is the time between ingestion of food and defecation and is considered one of the parameters to evaluate constipation, as it is longer with constipation [19]. GTT was significantly longer in the Lop group compared to that in the control group (Control: 4.4 ± 0.8 h, Lop: 6.1 ± 0.4 h). Further, GTT was significantly longer in the Lop + CPS group (6.9 ± 0.3 h; *p* < 0.05) than in the Lop group. In contrast, the Lop + SPS group (6.3 ± 0.5 h) was comparable to the Lop group, with no significant difference.

## 3. Discussion

The results of this study has demonstrated the lack of affect CPS had on the number of fecal pellets and the fecal wet weight but rather worsened constipation in the loperamide-induced constipation model by decreasing fecal water content and prolongating GTT. In contrast, SPS did not reduce fecal water content or increase GTT and improved the number of fecal pellets and the fecal wet weight; this suggests that SPS is a better alternative than CPS.

Constipation reduces the quality of life and complicates a variety of illnesses. In dialysis patients, who tend to experience constipation, this can be a major problem. The respective package inserts state that the incidence of constipation (as an adverse effect) is 9.2% for CPS (Package insert, kalimate powder. Kowa Pharmaceutical Co., Ltd. Japan. revised in April 2019) and 1.9% for SPS (Package insert, kayexalate powder. Torii Pharmaceutical Co., Ltd. Japan. Revised in January 2013). It has been previously reported that CPS causes constipation in approximately 8% of chronic kidney disease patients [20]. However, there has been no report comparing the frequency of constipation due to CPS and SPS factoring in other causes of constipation, such as the amount of food [21] and water [22].

When evaluating constipation from a defecation standpoint, the number of fecal pellets, fecal water content and GTT are used as general parameters. A decrease in the number of fecal pellets, a decrease in the fecal water content and a prolongation of GTT are considered common symptoms of constipation [19]. In particular, the decrease in the fecal water content and the prolongation of the GTT of the digestive tract are considered to be the main endpoints. The loperamide-induced constipation model is associated with decreased fecal water content, prolonged GTT and a decreased number of fecal pellets [23,24,25]. The results reported in the present study confirmed that constipation was induced in the loperamide-induced constipation model, due to a decrease in the number of fecal pellets (Figure 1), a decrease in fecal wet weight (Figure 2) and an increase in GTT (Figure 4). However, no significant decrease in the moisture content of the feces was observed (Figure 3). In the present study, based on past reports [26], we created a constipation model via the intraperitoneal administration of a loperamide suspension, but oral or subcutaneous administration is the common creation method of loperamide-induced constipation model. When a suspension is administered intraperitoneally, the absorption rate of the drug from the intraperitoneal cavity is more varied than with oral or subcutaneous administration. Therefore, we speculate that in the present study, the loperamide-induced constipation did not decrease the fecal water content, unlike that observed in the models established by oral or subcutaneous administration. However, the administration of CPS to the loperamide-induced constipation model further exacerbated changes in fecal water content (Figure 3) and GTT (Figure 4). Deterioration of the above parameters suggested that CPS exacerbated constipation. Conversely, SPS administration did not affect the fecal water content and GTT in loperamide-induced constipation model rats, but the number of fecal pellets (Figure 1) and fecal wet weight (Figure 2) were improved to the same extent as in the control group.

One of the causes of the difference in constipation exacerbation by CPS and SPS may be due to the release of Ca^2+^ and Na^+^ when K^+^ is adsorbed in the intestinal tract, respectively. CPS administration reduced the fecal water content in the loperamide-induced constipation model and significantly prolonged the GTT. Ca^2+^ released from CPS reacts with the bicarbonate and phosphoric acid in the intestinal tract to form insoluble salts such as calcium carbonate and calcium phosphate. As a result, the stool became hardened and caused a passage obstruction in the intestinal tract, thereby decreasing the water content in the feces. In fact, calcium carbonate itself has been reported to cause constipation [27]. On the other hand, SPS administration did not affect the fecal water content and GTT in the loperamide-induced constipation model but improved the number of fecal pellets and fecal wet weight to about the same level as the control group. It is expected that the Na^+^ released from SPS does not form an insoluble salt like Ca^2+^, but rather works to maintain the water content in the stool. Therefore, it is considered that the fecal water content and the GTT were not affected. However, this study has not sufficiently investigated the causes of constipation exacerbation due to CPS and SPS, and it is necessary to evaluate the electrolyte concentration in the intestinal tract and feces in the future and study it in more detail.

As shown in the present results, constipation may be avoided by preferentially administering SPS over CPS when dialysis patients present with hyperkalemia. Furthermore, this may help us avoid the fatal intestinal lesions such as intestinal obstructions and gastrointestinal perforations caused by exacerbation of constipation. This study is important in that it accurately evaluated a safe potassium binder in dialysis patients from the viewpoint of exacerbation of constipation using a constipation model.

In conclusion, we found that CPS did not affect the number of fecal pellets and the fecal wet weight, but rather, decreased fecal water content and prolongated GTT. Whereas SPS neither reduced fecal water content nor increased GTT, and rather improved the number of fecal pellets and the fecal wet weight. This suggested that SPS was less likely to cause exacerbation of constipation than CPS. Considering the high frequency of constipation in dialysis patients, it is considered that constipation exacerbation can be avoided by giving SPS priority over CPS, a potassium binder, when a patient presents with hyperkalemia.

## 4. Materials and Methods

### 4.1. Materials

CPS (Kalimate^®^ powder, Kowa Company Ltd., Tokyo, Japan) was purchased from Kowa Company, Ltd., SPS (Kayexalate^®^ powder) was procured from Torii Pharmaceutical Co, Ltd. (Tokyo, Japan) and normal saline was obtained from Fuso Pharmaceutical Industries, Ltd. (Osaka, Japan). Loperamide hydrochloride for pharmacological research was purchased from Wako Pure Chemical Industries, Ltd. (Osaka, Japan) and a guaranteed reagent grade carmine and liquid paraffin were purchased from Wako Pure Chemical Industries, Ltd.

### 4.2. Experimental Animals

Six to nine-weeks-old male Wistar rats (Kyudo Co, Ltd., Saga, Japan) were purchased and reared in a laboratory animal facility maintained at a room temperature of 24 ± 1 °C with a 12-h light–dark cycle (light hours 8:00 to 20:00). They were given free access to solid feed (CE-2, CLEA Japan, Inc., Tokyo, Japan); rats weighing 190 to 310 g were used in the experiment.

The animal experiments in this study were approved by the animal experiments committee of Kumamoto University (Honjo and Oe sites, approval number: B26–076, approval date: 1 April 2014) and were conducted in accordance with the Kumamoto University guidelines for animal experiments.

### 4.3. Preparation of Loperamide-Induced Constipation Model

The loperamide-induced constipation model was created by 3 days of twice-daily (9:00 and 18:00) intraperitoneal administration (ip) of loperamide hydrochloride (1 mg/kg body weight) suspended in physiological saline [26,28]. The control rats received ip injections of normal saline.

### 4.4. Experimental Protocol

The protocol of the crossover study is shown in Figure 5. A crossover study was conducted with eight rats. There were four treatment groups (*n* = 2 per group) as follows: control group, loperamide (Lop) group, loperamide plus CPS (Lop + CPS) group and loperamide plus SPS (Lop + SPS) group. The control group received normal feed, the Lop group received 1 mg/kg body weight of loperamide ip and the Lop + CPS and Lop + SPS groups received 1 mg/kg body weight of loperamide ip plus feed containing either 20% CPS or 20% SPS. After a 1-week washout period, a crossover was performed and the experiments in different groups were repeated four times. The experimental protocol for defecation observation is shown in Figure 6. After completion of the loperamide-induced constipation model, rats fasted for 24 h and were subsequently provided experimental feed dyed with 0.5% carmine for 1 h. Next, they fasted for 2 h and were given free access to undyed experimental feed. Defecation was assessed every hour for 18 h following administration. After 18 h of observation, a blood sample was collected from the tail vein. Electrolytes in the blood samples were assayed with the i-STAT^®^ 1 analyzer (Abbott Laboratories, Abbott Park, IL, USA). CPS or SPS was mixed into the feed and metabolic cages were used for the next 18 h for temporal observations. In addition, to maintain the drinking water restriction in dialysis patients, water intake was restricted by allowing each constipated rat only 40 mL/day of water.

### 4.5. Parameters Evaluated

In addition to the body weight tested at the end of the test, food intake, water intake and urine volume were also measured over the 18 h of the experiment, and the number of fecal pellets and fecal wet weight were measured each hour. Collected pellets were dried at room temperature for at least 48 h to measure the fecal dry weight, and the fecal water content was calculated by Equation (1). Further, GTT was considered as the time between consumption of feed dyed with 0.5% carmine to the time when red feces appeared [29].
(1)$$Fecal\;water\;content\;\left(\%\right)=\frac{\left(fecal\;wet\;weight\right)-\left(fecal\;dry\;weight\right)}{\left(fecal\;wet\;weight\right)}\times 100\;\;$$

### 4.6. Statistical Analysis

Experimental data were expressed as means ± standard deviation (SD). Statistical analysis was performed with Statcel4 (OMS publishing Inc., Saitama, Japan), an add-in software. Comparisons between groups were performed by one-way analysis of variance and significance testing by multiple comparison with Tukey’s (Tukey-Kramer) test. A result of less than 5% was considered statistically significant.

## Figures and Tables

**Figure 1 ijms-21-02491-f001:**
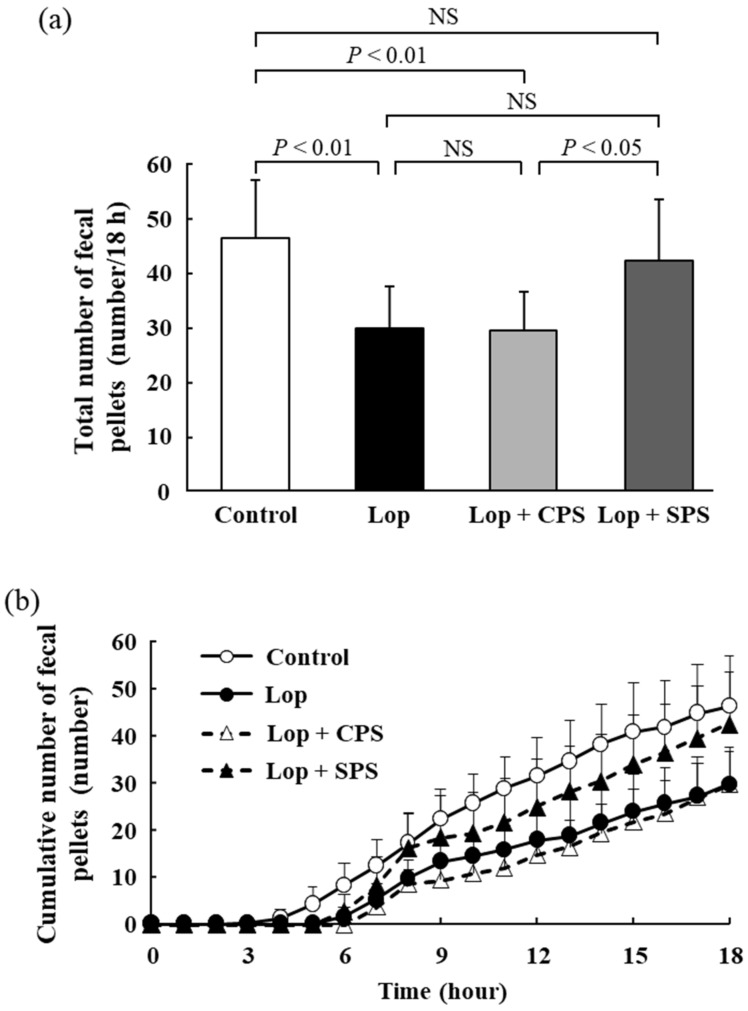
Effect of calcium polystyrene sulfonate (CPS) or sodium polystyrene sulfonate (SPS) on the number of fecal pellets. (**a**) total number of fecal pellets. (**b**) cumulative number of fecal pellets over time. Values are expressed as means ± S.D. (*n* = 8). Lop, loperamide-induced constipation; CPS, calcium polystyrene sulfonate; SPS, sodium polystyrene sulfonate; NS, not significant.

**Figure 2 ijms-21-02491-f002:**
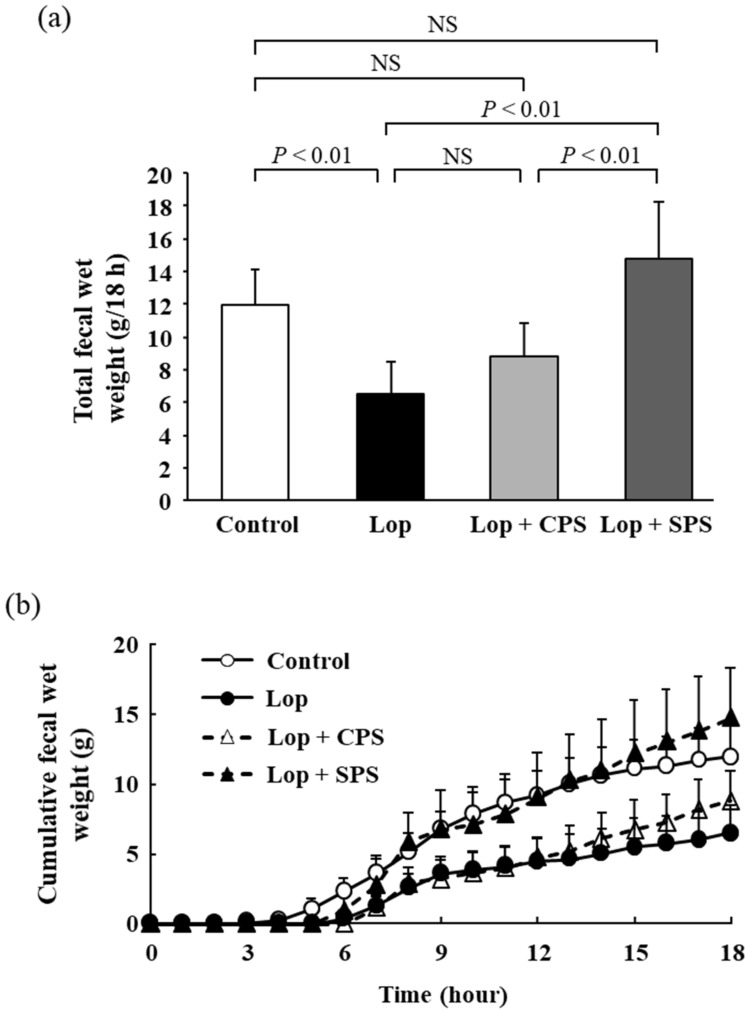
Effect of CPS or SPS on the fecal wet weight. (**a**) total fecal wet weight. (**b**) cumulative fecal wet weight over time. Values are expressed as means ± S.D. (*n* = 8). Lop, loperamide-induced constipation; CPS, calcium polystyrene sulfonate; SPS, sodium polystyrene sulfonate; NS, not significant.

**Figure 3 ijms-21-02491-f003:**
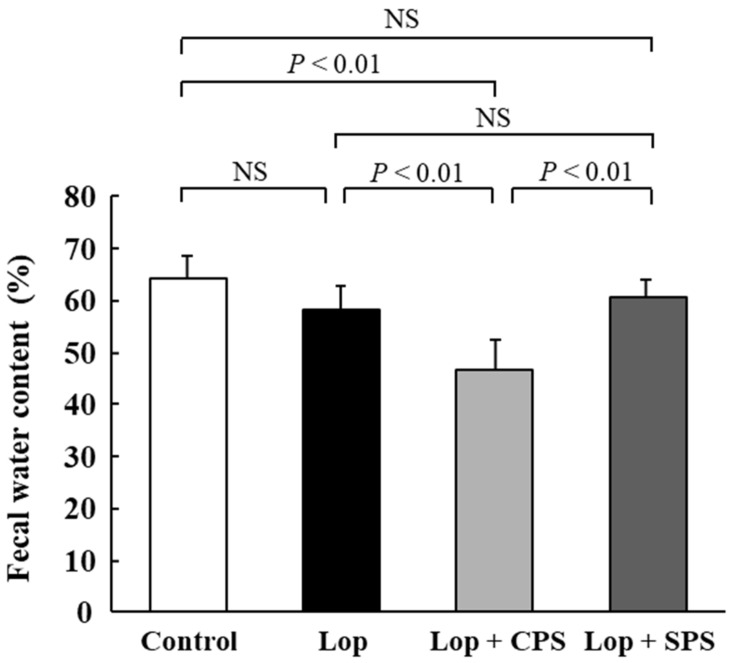
Effect of CPS or SPS on fecal water content. Fecal water content was calculated as ((fecal wet weight)—(fecal dry weight)/(fecal wet weight)) × 100. Values are expressed as means ± S.D. (*n* = 8). Lop, loperamide-induced constipation; CPS, calcium polystyrene sulfonate; SPS, sodium polystyrene sulfonate; NS, not significant.

**Figure 4 ijms-21-02491-f004:**
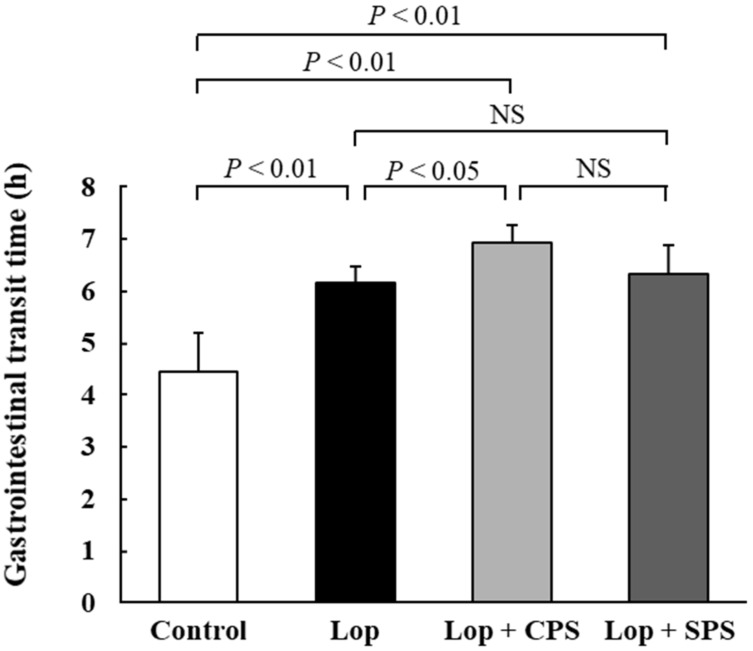
Effect of CPS or SPS on gastrointestinal transit time (GTT). GTT was considered as the time between consumption of feed dyed with 0.5% carmine to the time when red feces appeared. Values are expressed as means ± S.D. (*n* = 8). Lop, loperamide-induced constipation; CPS, calcium polystyrene sulfonate; SPS, sodium polystyrene sulfonate; NS, not significant.

**Figure 5 ijms-21-02491-f005:**
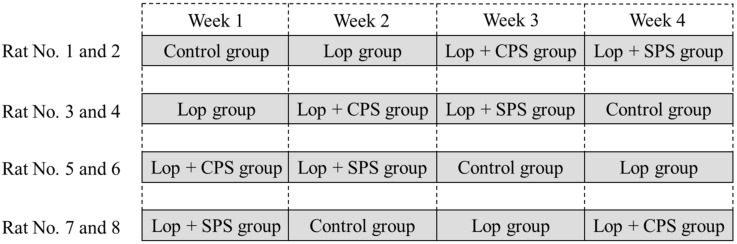
Crossover study protocol. A crossover study was performed with four treatment groups as follows: Control group, loperamide (Lop) group, loperamide plus CPS (Lop + CPS) group and loperamide plus SPS (Lop + SPS) group with two rats per group.

**Figure 6 ijms-21-02491-f006:**
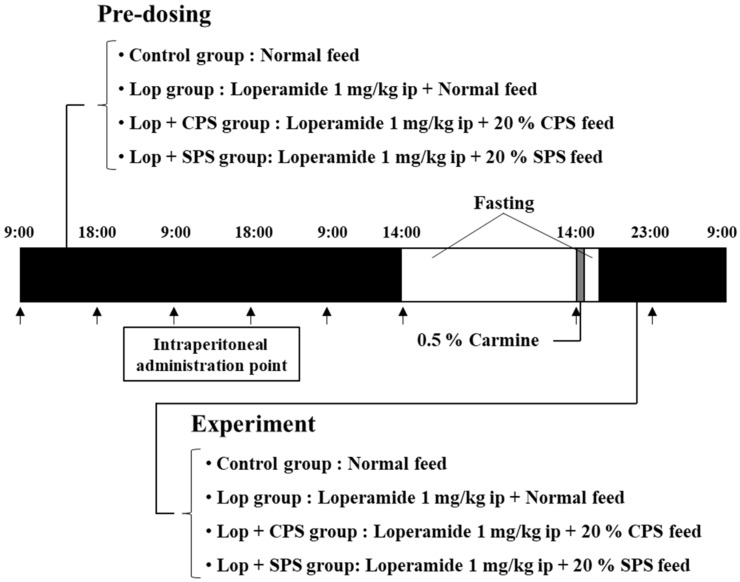
Experimental protocol. The loperamide-induced constipation model was created by the twice-daily (9:00 and 18:00) intraperitoneal administration (ip) of loperamide hydrochloride (1 mg/kg). After completion of the loperamide-induced constipation model, they fasted for 24 h and were subsequently provided experimental feed dyed with 0.5% carmine for 1 h. After a 2-h fast, defecation was assessed every hour for 18 h following administration. Water intake was restricted by allowing each constipated rat only 40 mL/day of water.

**Table 1 ijms-21-02491-t001:** Comparisons of various parameters among the four groups.

	Control	Lop	Lop + CPS	Lop + SPS	*p* value
N	8	8	8	8	
Body weight (g)	248.3 ± 45.8	248.1 ± 39.9	240.1 ± 35.0	254.1 ± 40.6	NS
Food intake (g/18 h)	20.4 ± 3.2	17.1 ± 3.1	20.7 ± 3.1	19.0 ± 1.7	NS
Water intake (mL/18 h)	36.4 ± 4.8	37.6 ± 3.8	37.9 ± 4.3	37.6 ± 3.2	NS
Urinary volume (mL/18 h)	19.4 ± 3.1	18.9 ± 2.9	16.1 ± 2.8	18.3 ± 2.9	NS

Values are expressed as means ± S.D. *p* < 0.05 indicates statistical significance. Lop, loperamide-induced constipation; CPS, calcium polystyrene sulfonate; SPS, sodium polystyrene sulfonate; NS, not significant.

**Table 2 ijms-21-02491-t002:** Comparisons of serum electrolytes among the four groups.

	Control	Lop	Lop + CPS	Lop + SPS	*p* value
N	8	8	8	8	
Na (mmol/L)	138.6 ± 4.0	140.8 ± 3.9	135.9 ± 8.5	142.0 ± 7.4	NS
K (mmol/L)	4.2 ± 0.6	4.1 ± 0.3	4.3 ± 0.8	3.6 ± 0.5	NS
Cl (mmol/L)	102.6 ± 2.0	103.0 ± 1.9	101.3 ± 3.1	102.6 ± 1.8	NS

Following 18 h of observation, a blood sample was collected from the tail vein and the electrolytes in the blood sample were assayed. Values are expressed as means ± S.D. *p* < 0.05 indicates statistical significance. Lop, loperamide-induced constipation; CPS, calcium polystyrene sulfonate; SPS, sodium polystyrene sulfonate; NS, not significant.

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
