# Peer review of "Comparative Study of Constipation Exacerbation by Potassium Binders Using a Loperamide-Induced Constipation Model"

_ijms, 2020, doi:10.3390/ijms21072491_

Round 1

Reviewer 1 Report

In this study Authors investigate two different potassium binders (CPS and SPS) in loperamide-induced constipation model in rats. Here are my specific comments:

Major comments:

1- The model authors use (loperamide) is actually a model of opioid-induced constipation, so I don't think it is relevant to ESRD-related constipation which authors discuss throughout the text.

2- The last paragraph of introduction is incorrect, loperamide model does not "mimic characteristics of dialysis patients with intestinal dysmotility", as stated above it mimics opioid-induced constipation.

3-The data in Figure 1 and 2 do not support the conclusions. From what I see CPS does not affect number of pellets or stool wet weight in loperamide-treated rats and SPS seems to be improving constipation. Thus there is no data in the study that supports the authors conclusion about CPS worsening constipation (other than some effect on water content and very small effect on GTT). Authors should comment on potential laxative effects of SPS. It is also essential to test SPS and CPS in control rats (not treated with loperamide) for their effects on stool parameters.

4- Figure 1 and 2: The statistical comparisons are confusing, I suggest using comparison brackets (descending from top) and showing comparisons between all groups. For instance in Fig 1A I wonder whether comparisons between Lop and Lop+SPS groups are significant, however this is not shown.

5-Figure 3: Interestingly loperamide itself does not decrease stool water content, this is definitely against many studies that show it does decrease both stool weight, number of pellets and water content. This makes me question the details of the experiment and authors report dissolving loperamide HCl in saline. The water solubility of loperamide HCl is very low (<0.001 mg/mL) thus I suppose either authors administered the loperamide despite it was not well-dissolved or they used huge volumes which would be against the animal treatment guidelines and also confound the constipation model.

6-First paragraph of the conclusion is not correct and not supported by data in the hand.

Minor comments:

1- Table 1 and 2: authors should explicitly state what NS means (not significant) and state the P-value threshold in the legend.

2- Figure 1: The titles for y-axis is confusing, it can be named as "Total number of pellets" for Figure 1a 

Author Response

Response to Reviewer 1 Comments

We wish to express our appreciation to the Reviewer 1 for the insightful comments, which have helped us significantly improve the manuscript.

Major comments:

Comment 1: The model authors use (loperamide) is actually a model of opioid-induced constipation, so I don't think it is relevant to ESRD-related constipation which authors discuss throughout the text.

Response 1: Thank you for your comment. Loperamide has been used in creating opioid-induced constipation animal models. Constipation in this model is caused by the suppression of intestinal motility by loperamide. Conversely, dialysis patients are prone to constipation primarily due to intestinal dysmotility (among other reasons) (Nishihara M et al., J Jpn Soc Dial Ther 2004, 37). Therefore, we considered that intestinal dysmotility could be caused by using loperamide, resulting in the loperamide-induced constipation model used in this study. In addition, we restricted water access and devised it so as to be close to the background of dialysis patients. Although it does not fully reflect all causes of constipation in dialysis patients, we considered the constipation model used in this study appropriate.

Comment 2: The last paragraph of introduction is incorrect, loperamide model does not "mimic characteristics of dialysis patients with intestinal dysmotility", as stated above it mimics opioid-induced constipation.

Response 2: Thank you for your comment. We believe that intestinal dysmotility, a primary cause of constipation in dialysis patients, can be accomplished by using loperamide. However, the description of "mimic characteristics of dialysis patients with intestinal dysmotility" may give the misunderstanding that the loperamide-induced constipation model reflects all causes of constipation in dialysis patients. So, we revised the text to "...reflecting intestinal dysmotility, which is a major cause of constipation in dialysis patients…”.

Comment 3: The data in Figure 1 and 2 do not support the conclusions. From what I see CPS does not affect number of pellets or stool wet weight in loperamide-treated rats and SPS seems to be improving constipation. Thus there is no data in the study that supports the authors conclusion about CPS worsening constipation (other than some effect on water content and very small effect on GTT). Authors should comment on potential laxative effects of SPS. It is also essential to test SPS and CPS in control rats (not treated with loperamide) for their effects on stool parameters.

Response 3: Thank you for your comment. CPS did not affect the number of fecal pellets and the fecal wet weight. However, symptoms of constipation in animals include not only low number of fecal pellets, but also low water content and slow intestinal transit rate (Liu J et al., Front Microbiol. 2018, 9, 3002). In addition, the primary cause of constipation is the long retention time of feces in the intestinal tract and the excessive absorption of water (Wang L et al., Int J Mol Sci. 2017, 18). In other words, reduction of fecal water content and prolongation of GTT can be the primary indicators. Thus, although CPS did not affect the number of fecal pellets and fecal wet weight, we believe that it exacerbated constipation in the loperamide-induced constipation model by decreasing fecal water content and prolongating GTT. But it may be misleading to state that CPS worsened all the indicators, therefore, we revised the conclusion to accurately reflected the results (Page 7, Line 190 - page 7, Line 193).

Additional Figures 1-4 show stool parameters when CPS and SPS were administered to Control rats. Administration of SPS to Control rats resulted in an increase in fecal wet weight and a decrease in fecal water content without affecting the number of fecal pellets and GTT. An increase in fecal wet weight has improved defecation, and a decrease in fecal water content has exacerbated it, resulting in conflicting results. Therefore, we cannot conclude that SPS has a potential laxative effect. We believe that the results obtained so far only suggest that SPS is less likely then CPS to cause constipation.

Additional Figure 1. Effect of CPS or SPS on the number of fecal pellets in control rats. Total number of fecal pellets. Values are expressed as means ± S.D. (n = 10).

Additional Figure 2. Effect of CPS or SPS on the fecal wet weight in control rats. Total fecal wet weight. Values are expressed as means ± S.D. (n = 10). **p < 0.01 compared with Control,p < 0.05 compared with CPS.

Additional Figure 3. Effect of CPS or SPS on fecal water content in control rats. Fecal water content was calculated as [(fecal wet weight) − (fecal dry weight) / (fecal wet weight) × 100]. Values are expressed as means ± S.D. (n = 10). *p < 0.05, **p < 0.01 compared with Control, p < 0.05 compared with Lop + CPS.

Additional Figure 4. Effect of CPS or SPS on gastrointestinal transit time (GTT) in control rats. GTT was considered as the time between consumption of feed dyed with 0.5% carmine to the time when red feces appeared. Values are expressed as means ± S.D. (n = 10).

Comment 4: Figure 1 and 2: The statistical comparisons are confusing, I suggest using comparison brackets (descending from top) and showing comparisons between all groups. For instance in Fig 1A I wonder whether comparisons between Lop and Lop+SPS groups are significant, however this is not shown.

Response 4: Thank you for your comment. Since the notation is mixed and statistical lack clarity, we have unified the notation in all Figures. In addition, we believe that more symbols used can be confusing, so we didn't marked insignificant differences. We hope that this revision will improve the clarity.

Comment 5: Figure 3: Interestingly loperamide itself does not decrease stool water content, this is definitely against many studies that show it does decrease both stool weight, number of pellets and water content. This makes me question the details of the experiment and authors report dissolving loperamide HCl in saline. The water solubility of loperamide HCl is very low (<0.001 mg/mL) thus I suppose either authors administered the loperamide despite it was not well-dissolved or they used huge volumes which would be against the animal treatment guidelines and also confound the constipation model.

Response 5: Thank you for your comment. The water solubility of loperamide HCl is very low and requires a large volume of solvent to completely dissolve. Therefore, a loperamide-induced constipation model was created by intraperitoneal administration of loperamide HCl suspended in physiological saline, as was previously reported (Kim JE et al., BMC Complement Altern Med. 2013, 13, 333.; Lee HY et al., Food Chem Toxicol 2012, 50, 895-902). We stated that "loperamide HCl dissolved in physiological saline” but revised the text to "loperamide HCl suspended in physiological saline". Therefore, we were within the animal treatment guidelines.

Comment 6: First paragraph of the conclusion is not correct and not supported by data in the hand.

Response 6: Thank you for your comment. It may be misleading to state that CPS exacerbated all of indicators, so we revised the conclusion to accurately reflect the results (Page 7, Line 190 - page 7, Line 193).

Minor comments:

Comment 1: Table 1 and 2: authors should explicitly state what NS means (not significant) and state the P-value threshold in the legend.

Response 1: Thank you for your comment. We revised the legend in table 1 and 2, explicitly stating that NS means not significant, and added a p-value threshold.

Comment 2: Figure 1: The titles for y-axis is confusing, it can be named as "Total number of pellets" for Figure 1a

Response 2: Thank you for your comment. We revised the y-axis title for Figure 1a to "Total number of fecal pellets" accordingly.

Reviewer 2 Report

I have carefully reviewed the manuscript entitled “Comparative study of constipation exacerbation by potassium binders using a loperamide-induced constipation model” (ijms-736612) by Yuki Narita et al.

In this comparative study, the authors used 32 rats (4 rats in each of the four groups: control; Lop; Lop+CPS; Lop+SPS) with loperamide-induced constipation model to evaluate the impact of potassium binders (CPS vs SPS) on constipation severity.

The authors found that SPS is less likely to cause constipation, comparing to CPS, in these rats, and suggested SPS is a preferred potassium lowering agent for in dialysis patients.

Generally speaking, this is a well-written manuscript.

The topic of the current study is practical and of clinical relevance.

The background, study design, data presentation and interpretation, as well as English writing are all good.

However, several concerns need to be addressed.

# Tables 1 & 2: Since many variables in the tables 1& 2 were not really effected by CPS/SPS (ex: food intake, water intake), I suggest changing table legends to “Comparisons of ### among four groups.”

# Table 2: What is the time point of obtaining these “serum electrolyte?” Please make a clear statement.

# 4.2.Experimental animals: I suggest the authors state the “approval number” of Ethic consideration.

# 4.4 Experimental protocol:I do not understand the meaning of the “crossover” in the study? How to cross over? Dose is mean “changing groups?” I suggest the authors make a more clear explanation in the protocol section.

Author Response

Comment 1: # Tables 1 & 2: Since many variables in the tables 1& 2 were not really effected by CPS/SPS (ex: food intake, water intake), I suggest changing table legends to “Comparisons of ### among four groups.”

Response 1: Thank you for your suggestion. We revised "Effects of CPS or SPS on various parameters" to "Comparisons of various parameters among the four groups" in Table 1 legend and "Effects of CPS or SPS on serum electrolytes" to "Comparisons of serum electrolytes among the four groups" in Table 2 legend.

Comment 2: # Table 2: What is the time point of obtaining these “serum electrolyte?” Please make a clear statement.

Response 2: Thank you for your comment. We have revised Table 2 legend and 4.4 Experimental protocol in the Methods section (Page 8, Line 228 - page 8, Line 230) regarding the points at which serum electrolytes were obtained.

Comment 3: # 4.2.Experimental animals: I suggest the authors state the “approval number” of Ethic consideration.

Response 3: Thank you for your suggestion. We added an approval number to 4.2. Experimental animals section of the revised manuscript (Page 7, Line 211).

Comment 4: # 4.4 Experimental protocol:I do not understand the meaning of the “crossover” in the study? How to cross over? Dose is mean “changing groups?” I suggest the authors make a more clear explanation in the protocol section.

Response 4: Thank you for your questions and suggestion. The crossover study means changing groups. We have rewritten the protocol section to be more in line with your comments (Page 8, Line 218 - page 8, Line 230), and added a new Figure 5 (Page 8) to help explain the crossover study.

Reviewer 3 Report

Authors performed an animal experiment to examine laxation associated with constipation induced animal using potassium binders which mimics dialysis patients using potassium binders in order to prevent hyperkalemia. The objective of the study may promote awareness of clinicians.

L97:       “Fecal wet weight has been reported to decrease during constipation [18].” This sentence is unneeded.

L112-114:            “Fecal water content is related to the hardness of stool and is considered as one of the parameters for evaluating constipation, as it decreases with constipation [18].”  Unnecessary sentence.

L142-149:            Authors describe troubles caused by constipation among people including healthy persons. This manuscript compares effect of potassium binders on constipation, and these agents are used only for severe CKD patients with hyperkalema. Description on healthy people should not be included.

L144-145:            “The respective package inserts state that the incidence of constipation as an adverse reaction is 9.2% for CPS and 1.9% for SPS.”

The sentence is unclear and needs a reference.

Methods section,

How and when were the blood specimens obtained?

Author Response

Comment 1: L97:       “Fecal wet weight has been reported to decrease during constipation [18].” This sentence is unneeded.

Response 1: Thank you for your comment. We have removed “Fecal wet weight has been reported to decrease during constipation [18].”

Comment 2: L112-114:            “Fecal water content is related to the hardness of stool and is considered as one of the parameters for evaluating constipation, as it decreases with constipation [18].”  Unnecessary sentence.

Response 2: Thank you for your comment. We removed “Fecal water content is related to the hardness of stool and is considered as one of the parameters for evaluating constipation, as it decreases with constipation [18].”

Comment 3: L142-149:            Authors describe troubles caused by constipation among people including healthy persons. This manuscript compares effect of potassium binders on constipation, and these agents are used only for severe CKD patients with hyperkalema. Description on healthy people should not be included.

Response 3: Thank you for your comment. The indication for CPS and SPS is hyperkalemia associated with acute and chronic kidney disease. Therefore, the description regarding healthy individuals was deleted and the text has been revised accordingly (Page 6, Line 148 - page 6, Line 153).

Comment 4: L144-145:            “The respective package inserts state that the incidence of constipation as an adverse reaction is 9.2% for CPS and 1.9% for SPS.” The sentence is unclear and needs a reference.

Response 4: Thank you for your comment. We added the relevant reference to the statement regarding constipation as an adverse reaction in Discussion section of the revised manuscript (Page 6, Line 148 - page 6, Line 152).

Comment 5: (Methods section) How and when were the blood specimens obtained?

Response 5: Thank you for your question. We have revised section 4.4 Experimental protocol in the Methods to address this (Page 8, Line 228 - page 8, Line 230).

Round 2

Reviewer 1 Report

Authors state that they administered loperamide as a suspension via intraperitoneal route, which is not acceptable from an experimental design stand point. In experimental animals and also in clinic parenteral treatments are almost always clear solutions and administering suspension in to peritoneal cavity will cause a lot of variability in terms of compound absorption to the blood stream. Thus it is not surprising that loperamide did not cause a decrease in stool water content in this study, and I speculate that it was due to erroneous experimental design. In addition the reference authors cite (Kim JE et al., BMC Complement Altern Med2013, 13, 333) actually administers loperamide subcutaneously and there is no mention aof suspension in that study. 

Although authors state that they included 4 Additional Figures, they are not present among the submission documents. In addition they do not discuss those results or refer to the figures in the text. Thus it is hard to say that the new figures really exist.

Figure 1 and 2: The statistical comparisons are still confusing, I strongly suggest using comparison brackets (descending from top) and showing comparisons between all groups.

Author Response

Response: Thank you for your comment. Common methods to create a loperamide-induced constipation model are via oral or subcutaneous administration . However, there is one report of a loperamide-induced constipation model prepared by intraperitoneally administering a loperamide suspension (Lee HY et al., Food Chem Toxicol 2012, 50, 895-902), in which the abstract and methods includes the description "Loperamide (2 mg / kg, twice per day) was injected intraperitoneally to induce constipation in the four experimental groups" and “Following 4 weeks of FP intake, we injected rats with loperamide (Lopmin Cap, Youngilpharm Co., Korea, 2 mg/kg) suspended in 0.9% sodium chloride twice per day at 09:00 and at 18:00 h for 1 week”. From this description, we deciphered that constipation can be induced via the intraperitoneal administration of loperamide suspension. However, as stated in your comment, when a suspension is administered intraperitoneally, the absorption rate of the drug from the intraperitoneal cavity is more variable than that with oral administration or subcutaneous administration. Therefore, it is suggested in the study outcomes that the loperamide-induced constipation model in this study did not decrease the fecal water content unlike that observed in the models prepared via oral or subcutaneous administration. Regarding these points, we have revised the discussion (Page 7, Line 162-175). Moreover, as you pointed out, the report by Kim et al did not include a detailed description of the loperamide suspension, and the administration method was different, and thus, we have removed it from the references. We again thank you for pointing this out.

Response: The Additional Figure disappeared due to an unknown error. Accordingly, please see our response as follows. Additional Figures 1–4 show stool parameters when CPS and SPS were
administered to Control rats. The total number of fecal pellets is shown in Additional Figure 1. The total number of fecal pellets was not significantly different between the Control group (63.1 ± 10.7 pellets/18 h) and the Lop + SPS group (69.4 ± 6.8 pellets/18 h). The total fecal wet weight is shown in Additional Figure 2. The total fecal wet weight was significantly heavier in the Lop + SPS group (23.6 ± 2.5 g/18 h) than in the Control group (17.6 ± 2.9 g/18 h). Fecal water content is shown in Additional Figure 3. Fecal water content was significantly lower in the Lop + SPS group (61.4% ± 2.4%) than in the Control group (66.1% ± 3.1 %). GTT is shown in Additional Figure 4. GTT was not significantly different between the Control group (4.6 ± 0.7 h) and the Lop + SPS group (5.2 ± 0.6 h). These results showed that the administration of SPS to Control rats resulted in an increase in fecal wet weight and a decrease in fecal water content without affecting the number of fecal pellets and GTT. Since an increase in fecal wet weight would improve defecation, and a decrease in fecal water content would exacerbate it, these results were conflicting. Therefore, based on these results, we cannot conclude that SPS has a potential laxative effect. We believe that the results obtained only suggest that SPS is less likely then CPS to cause constipation.
Additional Figure 1. Effect of calcium polystyrene sulfonate (CPS) or sodium polystyrene sulfonate (SPS) on the number of fecal pellets in control rats. Total number of fecal pellets. Values are expressed as means ± S.D. (n = 10).

Additional Figure 2. Effect of calcium polystyrene sulfonate (CPS) or sodium polystyrene sulfonate (SPS) on the fecal wet weight in control rats. Total fecal wet weight. Values are expressed as means ± S.D. (n = 10).

Additional Figure 3. Effect of calcium polystyrene sulfonate (CPS) or sodium polystyrene sulfonate (SPS) on fecal water content in control rats. Fecal water content was calculated as [(fecal wet weight) − (fecal dry weight) / (fecal wet weight) × 100]. Values are expressed as means ± S.D. (n = 10).

Additional Figure 4. Effect of calcium polystyrene sulfonate (CPS) or sodium polystyrene sulfonate (SPS) on gastrointestinal transit time (GTT) in control rats. GTT was considered the time between consumption of feed dyed with 0.5% carmine to the time when red feces appeared. Values are expressed as means ± S.D. (n = 10).

Response: Thank you for your suggestion. We modified the statistical comparisons using comparison brackets in all figures to show comparisons among all groups.